# Counterfactual Graph Learning for Link Prediction

## Abstract

Learning to predict missing links is important for many graph-based applications. Existing methods were designed to learn the observed association between two sets of variables: (1) the observed graph structure and (2) the existence of link between a pair of nodes. However, the causal relationship between these variables was ignored and we visit the possibility of learning it by simply asking a counterfactual question: "*would the link exist or not if the observed graph structure became different?*" To answer this question by causal inference, we consider the information of the node pair as context, global graph structural properties as treatment, and link existence as outcome. In this work, we propose a novel link prediction method that enhances graph learning by the counterfactual inference. It creates counterfactual links from the observed ones, and our method learns representations from both of them. Experiments on a number of benchmark datasets show that our proposed method achieves the state-of-the-art performance on link prediction.

## 1   Introduction

Link prediction seeks to predict the likelihood of edge existence between node pairs based on the observed graph. Given the omnipresence of graph-structured data, link prediction has copious applications such as movie recommendation (Bennett et al., 2007), chemical interaction prediction (Stanfield et al., 2017), and knowledge graph completion (Kazemi and Poole, 2018). Graph machine learning methods have been widely applied to solve this problem. Their standard scheme is to first learn the representation vectors of nodes and then learn the *association* between the representations of a pair of nodes and the existence of the link between them. For example, graph neural networks (GNNs) use neighborhood aggregation to create the representation vectors: the representation vector of a node is computed by recursively aggregating and transforming representation vectors of its neighboring nodes (Kipf and Welling, 2016a; Hamilton et al., 2017; Wu et al., 2020). Then the vectors are fed into a binary classification model to learn the *association*. GNN methods have shown predominance in the task of link prediction (Kipf and Welling, 2016b; Zhang and Chen, 2018; Zhang et al., 2020a).

Unfortunately, the causal relationship between graph structure and link existence was largely ignored in the previous work. Existing methods that learn from association only were not able to capture essential factors to accurately predict missing links in the *test data*. Take social network as an example. Suppose Alice and Adam live in the same neighborhood and they are close friends. The association between neighborhood belonging and friend closeness could be too strong to discover the essential factors of the friendship such as common interests or family relationship which could be the cause of being living in the same neighborhood. So, our idea is to ask a *counterfactual* question: "*would Alice and Adam still be close friends if they were not living in the same neighborhood?*" If a graph learning model could learn the causal relationship from data by asking the counterfactual questions, it would improve the performance of link prediction with the novel knowledge it captured. Generally, the questions can be described as "*would the link exist or not if the graph structure became different?*"

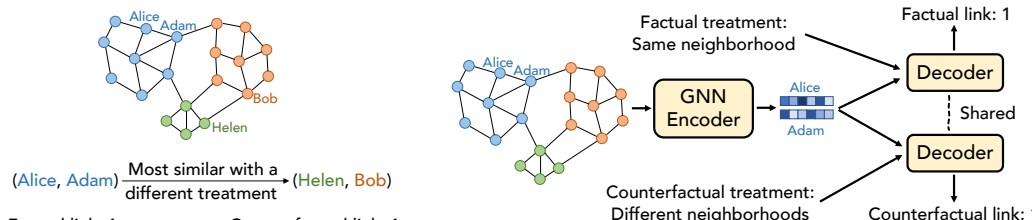

(a) Find counterfactual link as the most similar node pair with a different treatment.

(b) Train a GNN-based link predictor to predict factual and counterfactual links given the corresponding treatments.

Figure 1: The proposed CFLP learns the causal relationship between the observed graph structure (e.g., neighborhood similarity, considered as treatment variable) and link existence (considered as outcome). In this example, the link predictor would be trained to estimate the individual treatment effect (ITE) as $1 - 1 = 0$ so it looks for factors other than neighborhood to predict the factual link.

As known to many, counterfactual question is a key component of causal inference and have been well defined in the literature. A counterfactual question is usually framed with three factors: context (as a data point), manipulation (e.g., treatment, intervention, action, strategy), and outcome (van der Laan and Petersen, 2007; Johansson et al., 2016). (To simplify the language, we use "treatment" to refer to the manipulation in this paper, as readers might be familiar more with the word "treatment.") Given certain data context, it asks what the outcome would have been if the treatment had not been the observed value. In the scenario of link prediction, we consider the information of a pair of nodes as context, graph structural properties as treatment, and link existence as outcome. Recall the social network example. The context is Alice and Adam, which includes their personal attributes and relationships with others on the network. The treatment is living in the same neighborhood, which can be given as one attribute or identified by community detection. And the outcome is their friendship.

In this work, we present a **c**ounter**f**actual graph learning method for **l**ink **p**rediction (CFLP) that trains graph learning models to answer the counterfactual questions. Figure 1 illustrates this two-step method. Suppose the treatment variable is defined as one type of global graph structure, e.g., the neighborhood assignment discovered by spectral clustering or community detection algorithms. We are wondering how likely the neighborhood distribution makes a difference on the link (non-)existence for each pair of nodes. So, given a pair of nodes (like Alice and Adam) and the treatment value on this pair (in the same neighborhood), we find a pair of nodes (like Helen and Bob) that satisfies two conditions: (1) it has a different treatment (in different neighborhoods) and (2) it is the most similar pair with the given pair of nodes. We call these matched pair of nodes as "counterfactual links." Note that the outcome of the counterfactual link can be either 1 or 0, depending on whether there exists an edge between the matched pair of nodes. The counterfactual link provides unobserved outcome to the given pair of nodes (Alice and Adam) under a counterfactual condition (in different neighborhoods). After counterfactual links are created for all (positive and negative) training examples, CFLP trains a link predictor (which can be GNN-based) to learn the representation vectors of nodes to predict both the observed factual links and the counterfactual links. In this Alice-Adam example, the link predictor is trained to estimate the individual treatment effect (ITE) of neighborhood assignment as $1 - 1 = 0$. So, the learner will try to discover the essential factors on the friendship between Alice and Adam. For some other examples, if the outcome of counterfactual link is different from that of the given pair of nodes, the learner will estimate the strong effect of the treatment variable. Therefore, CFLP enables graph learning models to predict missing links regarding causal relationship.

**Contributions.** Our main contributions can be summarized as follows. (1) This is the first work that proposes to improve link prediction by causal inference, specifically, learning to answer counterfactual questions about link existence. (2) This work introduces CFLP that trains GNN-based link predictors to predict both factual and counterfactual links. It learns the causal relationship between global graph structure and link existence. (3) CFLP outperforms competitive baseline methods on several benchmark datasets. On OGB-DDI, our CFLP achieves the state-of-the-art performance. We analyze the impact of counterfactual links as well as the choice of treatment variable. This work sheds insights for improving graph machine learning with causal analysis, which has not been extensively studied yet, when the other direction (machine learning for causal inference) has been studied for a long time.

## 2 Preliminary

**Notations** Let $G = (\mathcal{V}, \mathcal{E})$ be an undirected graph of $N$ nodes, where $\mathcal{V} = \{v_1, v_2, \ldots, v_N\}$ is the set of nodes and $\mathcal{E} \subseteq \mathcal{V} \times \mathcal{V}$ is the set of observed links. We denote the adjacency matrix as $\mathbf{A} \in \{0, 1\}^{N \times N}$, where $A_{i,j} = 1$ indicates nodes $v_i$ and $v_j$ are connected and vice versa. We denote the node feature matrix as $\mathbf{X} \in \mathbb{R}^{N \times F}$, where $F$ is the number of node features and $\mathbf{x}_i$ (bolded) indicates the feature vector of node $v_i$ (the $i$-th row of $\mathbf{X}$).

**Counterfactual Learning** Let $\mathcal{X}$ be the set of contexts, $\mathcal{Y}$ be the set of outcome values, and $\mathcal{T}$ be the set of treatments. For a context $x \in \mathcal{X}$ and a treatment $t \in \mathcal{T}$, we denote the outcome of $x$ under the treatment $t$ by $Y_t(x) \in \mathcal{Y}$. Ideally, we would need all possible outcomes of $x$ under all kinds of treatments to study the causal relationships (Morgan and Winship, 2015). However, in reality, only one treatment was applied and thus only one outcome was observed for a given context $x$. When the variables are specified in data, people use Neyman–Rubin casual model (BCM) to develop statistical learning methods such as propensity score matching (PSM) for causal inference (Rubin, 1974, 2005).

In this work, we look at **link prediction** on graphs. Here we define the variables of counterfactual learning in this scenario. Given a graph $G$, a context is a pair of nodes $x = (v_i, v_j)$ in the graph; and thus, $\mathcal{X} = \mathcal{V} \times \mathcal{V}$. The outcome variable $Y(x)$ is naturally binary, indicating whether a link exists between the node pair $x$; and thus, $\mathcal{Y} = \{0, 1\}$. We study the causal effect of binary treatment variable $t \in \mathcal{T} = \{0, 1\}$, where the value of $Y_1(x) - Y_0(x)$ for a particular context $x$ is of high interest and known as the *individualized treatment effect* (ITE) (van der Laan and Petersen, 2007; Weiss et al., 2015). The value of ITE indicates the causality relationship between the treatment and outcome on the context. And the expected ITE given the context distribution is called *averaged treatment effect* (ATE). i.e., ATE $= \mathbb{E}_{x \sim \mathcal{X}} \text{ITE}(x)$, for a particular treatment variable.

However, as aforementioned, the fact that we can only observe one potential outcome under one particular treatment prevents the ITE from being known (Johansson et al., 2016). In the problem setting of link prediction, we refer the observed adjacency matrix as the *factual* outcomes $\mathbf{A}$ and the unobserved adjacency matrix when the treatment is different as the *counterfactual* outcomes $\mathbf{A}^{CF}$. We denote $\mathbf{T} \in \{0, 1\}^{N \times N}$ as the factual treatment matrix, where $T_{i,j}$ indicates the treatment of the node pair $(v_i, v_j)$. We denote $\mathbf{T}^{CF}$ as the counterfactual treatment matrix where $T_{i,j}^{CF} = 1 - T_{i,j}$. We are interested in (1) estimating the counterfactual outcomes $\mathbf{A}^{CF}$ via observed data, (2) learning with the counterfactual adjacency matrix $\mathbf{A}^{CF}$ to enhance link prediction, and (3) learning the causal relationship between graph structural information (treatment) and link existence (outcome).

## 3 The Proposed Method

In this section, we introduce CFLP, a novel counterfactual graph learning method for link prediction. In Section 3.1, we define treatment variable and counterfactual outcomes/links on graph data and present how to compute them (Figure 1(a)). In Section 3.2, we introduce the graph learning model that learns from both the observed graph and the created counterfactual links (Figure 1(b)).

### 3.1 Defining Treatment Variable and Counterfactual Links

**Treatment** Previous work on graph machine learning (Velickovic et al., 2019; Park et al., 2020) showed that the graph's global structural information could improve the quality of representation vectors of nodes learned by GNNs. This is because the message passing-based GNNs aggregate local information in the algorithm of representation vector generation and the global structural information is complementary with the aggregated information. Therefore, for a pair of nodes, one option of defining the treatment variable is its global structural role in the graph. Without the loss of generality, we use Louvain (Blondel et al., 2008), an unsupervised approach that has been widely used for community detection, as an example. Louvain discovers community structure of a graph and assigns each node to one community. Then we can define the binary treatment variable as whether these two nodes in the pair belong to the same community. Let $c : \mathcal{V} \to \mathbb{N}$ be any graph mining/clustering method that outputs the index of community/cluster/neighborhood that each node belongs to. The treatment matrix $\mathbf{T}$ is defined as

$$T_{i,j} = \begin{cases} 1 & \text{, if } c(v_i) = c(v_j); \\ 0 & \text{, otherwise.} \end{cases} \tag{1}$$

127 For the choice of $c$, we suggest methods that group nodes based on global graph structural information,
128 including but not limited to Louvain (Blondel et al., 2008), K-core (Bader and Hogue, 2003), and
129 spectral clustering (Ng et al., 2001).

130 **Counterfactual Links**    As mentioned in Section 2, for each node pair (context), the observed data
131 contains only the factual treatment and outcome, meaning that the link existence for the given node
132 pair with an opposite treatment is unknown. Therefore, we use the outcome from the nearest observed
133 context as a substitute. This idea has been adopted by many methods (Johansson et al., 2016; Alaa and
134 Van Der Schaar, 2019). That is, we want to find the nearest neighbor with the opposite treatment for
135 each observed node pairs and use the nearest neighbor's outcome as a *counterfactual link*. Formally,
136 $\forall (v_i, v_j) \in \mathcal{V} \times \mathcal{V}$, we want to find its counterfactual link $(v_a, v_b)$ as below:

$$(v_a, v_b) = \underset{v_a, v_b \in \mathcal{V}}{\arg\min} \{ d((v_i, v_j), (v_a, v_b)) \mid T_{a,b} = 1 - T_{i,j} \}, \tag{2}$$

137 where $d(\cdot, \cdot)$ is a metric of measuring the distance between a pair of node pairs (a pair of contexts).
138 Considering that we want to find the nearest node pair based on not only the raw node features but
139 also structural features, here we take the state-of-the-art unsupervised graph representation learning
140 method MVGRL (Hassani and Khasahmadi, 2020) to learn the node embeddings $\tilde{\mathbf{X}} \in \mathbb{R}^{N \times \tilde{F}}$ from
141 the observed graph. We use $\tilde{\mathbf{X}}$ to find the nearest neighbors of node pairs. Nevertheless, finding the
142 nearest neighbors by computing the distance between all pairs of node pairs is extremely inefficient,
143 which takes $O(N^4)$ comparisons (as there are totally $O(N^2)$ node pairs). Hence we approximate
144 Eq. (2) by substituting the distance between node pairs by the distance between nodes. That is,
145 $\forall (v_i, v_j) \in \mathcal{V} \times \mathcal{V}$, we want to find its counterfactual link $(v_a, v_b)$ as below:

$$(v_a, v_b) = \underset{v_a, v_b \in \mathcal{V}}{\arg\min} \{ d(\tilde{\mathbf{x}}_i, \tilde{\mathbf{x}}_a) + d(\tilde{\mathbf{x}}_j, \tilde{\mathbf{x}}_b) \mid T_{a,b} = 1 - T_{i,j}, d(\tilde{\mathbf{x}}_i, \tilde{\mathbf{x}}_a) + d(\tilde{\mathbf{x}}_j, \tilde{\mathbf{x}}_b) < 2\gamma \}, \tag{3}$$

146 where $d(\cdot, \cdot)$ is specified as the Euclidean distance on the embedding space of $\tilde{\mathbf{X}}$, and $\gamma$ is a hyperpa-
147 rameter that defines the maximum distance that two nodes are considered as similar. Note that when
148 no node pair satisfies the above equation, we do not assign any nearest neighbor for a given node pair
149 to ensure all the neighbors are similar enough (as substitutes) in the feature space. Therefore, the
150 counterfactual treatment matrix $\mathbf{T}^{CF}$ and the counterfactual adjacency matrix $\mathbf{A}^{CF}$ are defined as

$$T_{i,j}^{CF}, A_{i,j}^{CF} = \begin{cases} 1 - T_{i,j}, A_{a,b} & \text{, if } \exists\, (v_a, v_b) \in \mathcal{V} \times \mathcal{V} \text{ satisfies Eq. (3);} \\ T_{i,j}, A_{i,j} & \text{, otherwise.} \end{cases} \tag{4}$$

151 It is worth noting that the node embeddings $\tilde{\mathbf{X}}$ and the nearest neighbors are computed only once and
152 do not change during the learning process. $\tilde{\mathbf{X}}$ is only used for finding the nearest neighbors.

153 **Learning from Counterfactual Distributions**    Let $P^F$ be the factual distribution of the observed
154 contexts and treatments, and $P^{CF}$ be the counterfactual distribution that is composed of the observed
155 contexts and opposite treatments. We define the empirical factual distribution $\hat{P}^F \sim P^F$ as $\hat{P}^F =$
156 $\{(v_i, v_j, T_{i,j}^F)\}_{i,j=1}^N$, and define the empirical counterfactual distribution $\hat{P}^{CF} \sim P^{CF}$ as $\hat{P}^{CF} =$
157 $\{(v_i, v_j, T_{i,j}^{CF})\}_{i,j=1}^N$. Unlike traditional link prediction methods that take only $\hat{P}^F$ as input and use
158 the observed outcomes $\mathbf{A}$ as the training target, the idea of counterfactual graph learning is to take
159 advantage of the counterfactual distribution by having $\hat{P}^{CF}$ as a complementary input and use the
160 counterfactual outcomes $\mathbf{A}^{CF}$ as the training target for the counterfactual data samples.

## 3.2   The Counterfactual Graph Learning Model

162 In this subsection, we present the design of our model as well as the training method. The input of
163 the model in CFLP includes (1) the observed graph data $\mathbf{A}$ and raw feature matrix $\mathbf{X}$, (2) the factual
164 treatments $\mathbf{T}^F$ and counterfactual treatments $\mathbf{T}^{CF}$, and (3) the counterfactual graph data $\mathbf{A}^{CF}$. The
165 output contains link prediction logits in $\widehat{\mathbf{A}}$ and $\widehat{\mathbf{A}}^{CF}$ for the factual and counterfactual adjacency
166 matrices $\mathbf{A}$ and $\mathbf{A}^{CF}$, respectively.

167 **Graph Learning Model**    The model consist of two trainable components: a graph encoder $f$ and a
168 link decoder $g$. The graph encoder generates representation vectors of nodes from graph data $G$. And
169 the link decoder projects the representation vectors of node pairs into the link prediction logits. The

choice of the graph encoder $f$ can be any end-to-end GNN model. Without the loss of generality, here we use the commonly used graph convolutional network (GCN) (Kipf and Welling, 2016a). Each layer of GCN is defined as

$$\mathbf{H}^{(l)} = f^{(l)}(\mathbf{A}, \mathbf{H}^{(l-1)}; \mathbf{W}^{(l)}) = \sigma(\tilde{\mathbf{D}}^{-\frac{1}{2}}\tilde{\mathbf{A}}\tilde{\mathbf{D}}^{-\frac{1}{2}}\mathbf{H}^{(l-1)}\mathbf{W}^{(l)}), \tag{5}$$

where $l$ is the layer index, $\tilde{\mathbf{A}} = \mathbf{A} + \mathbf{I}$ is the adjacency matrix with added self-loops, $\tilde{\mathbf{D}}$ is the diagonal degree matrix $\tilde{D}_{ii} = \sum_j \tilde{A}_{ij}$, $\mathbf{H}^{(0)} = \mathbf{X}$, $\mathbf{W}^{(l)}$ is the learnable weight matrix at the $l$-th layer, and $\sigma(\cdot)$ denotes a nonlinear activation such as ReLU. We denote $\mathbf{Z} = f(\mathbf{A}, \mathbf{X}) \in \mathbb{R}^{N \times H}$ as the output from the encoder's last layer, i.e., the $H$-dimensional representation vectors of nodes. Following previous work (Zhang et al., 2020a), we compute the representation of a node pair as the Hadamard product of the vectors of the two nodes. That is, the representation for the node pair $(v_i, v_j)$ is $\mathbf{z}_i \odot \mathbf{z}_j \in \mathbb{R}^H$, where $\odot$ stands for the Hadamard product.

For the link decoder that predicts whether a link exists between a pair of nodes, we opt for simplicity and adopt a simple decoder based on multi-layer perceptron (MLP), given the representations of node pairs and their treatments. That is, the decoder $g$ is defined as

$$\widehat{\mathbf{A}} = g(\mathbf{Z}, \mathbf{T}), \text{ where } \widehat{A}_{i,j} = \text{MLP}([\mathbf{z}_i \odot \mathbf{z}_j, T_{i,j}]), \tag{6}$$

$$\widehat{\mathbf{A}}^{CF} = g(\mathbf{Z}, \mathbf{T}^{CF}), \text{ where } \widehat{A}_{i,j}^{CF} = \text{MLP}([\mathbf{z}_i \odot \mathbf{z}_j, T_{i,j}^{CF}]), \tag{7}$$

where $[\cdot, \cdot]$ stands for the concatenation of vectors.

During the training process, data samples from the empirical factual distribution $\hat{P}^F$ and the empirical counterfactual distribution $\hat{P}^{CF}$ are fed into decoder $g$ and optimized towards $\mathbf{A}$ and $\mathbf{A}^{CF}$, respectively. That is, for the two distributions, the loss functions are as follows:

$$\mathcal{L}_F = \frac{1}{N^2} \sum_{i=1}^{N} \sum_{j=1}^{N} A_{i,j} \cdot \log \widehat{A}_{i,j} + (1 - A_{i,j}) \cdot \log(1 - \widehat{A}_{i,j}), \tag{8}$$

$$\mathcal{L}_{CF} = \frac{1}{N^2} \sum_{i=1}^{N} \sum_{j=1}^{N} A_{i,j}^{CF} \cdot \log \widehat{A}_{i,j}^{CF} + (1 - A_{i,j}^{CF}) \cdot \log(1 - \widehat{A}_{i,j}^{CF}). \tag{9}$$

**Balancing Counterfactual Learning**    In the training process, the above loss minimizations train the model on both the empirical factual distribution $\hat{P}^F \sim P^F$ and empirical counterfactual distribution $\hat{P}^{CF} \sim P^{CF}$ that are not necessarily equal – the training examples (node pairs) do not have to be aligned. However, at the stage of inference, the test data contains only observed (factual) samples. Such a gap between the training and test data distributions exposes the model in the risk of covariant shift, which is a common issue in counterfactual learning (Johansson et al., 2016; Assaad et al., 2021).

To force the distributions of representations of factual distributions and counterfactual distributions to be similar, we use the discrepancy distance (Mansour et al., 2009; Johansson et al., 2016) as another objective to regularize the representation learning. That is, we use the following loss term to minimize the distance between the learned representations from $\hat{P}^F$ and $\hat{P}^{CF}$:

$$\mathcal{L}_{disc} = \text{disc}(\hat{P}_f^F, \hat{P}_f^{CF}), \text{ where } \text{disc}(P, Q) = ||P - Q||_F, \tag{10}$$

where $|| \cdot ||_F$ denotes the Frobenius Norm, and $\hat{P}_f^F$ and $\hat{P}_f^{CF}$ denote the node pair representations learned by graph encoder $f$ from factual distribution and counterfactual distribution, respectively.

**Training**    During the training of CFLP, we want the model to be optimized towards three targets: (1) accurate link prediction on the observed outcomes (Eq. (8)), (2) accurate estimation on the counterfactual outcomes (Eq. (9)), and (3) regularization on the representation spaces learned from $\hat{P}^F$ and $\hat{P}^{CF}$ (Eq. (10)). Therefore, the overall training loss of our proposed CFLP is

$$\mathcal{L} = \mathcal{L}_F + \alpha \cdot \mathcal{L}_{CF} + \beta \cdot \mathcal{L}_{disc}, \tag{11}$$

where $\alpha$ and $\beta$ are hyperparameters to control the weights of counterfactual link prediction (outcome estimation) loss and discrepancy loss.

Table 1: Statistics of datasets used in the experiments.

| Dataset | CORA | CITESEER | PUBMED | FACEBOOK | OGB-DDI |
|---|---|---|---|---|---|
| # nodes | 2,708 | 3,327 | 19,717 | 4,039 | 4,267 |
| # links | 5,278 | 4,552 | 44,324 | 88,234 | 1,334,889 |
| # validation node pairs | 1,054 | 910 | 8,864 | 17,646 | 235,371 |
| # test node pairs | 2,110 | 1,820 | 17,728 | 35,292 | 229,088 |

**Summary**  Algorithm 1 summarizes the whole process of CFLP. The **first step** is to compute the factual and counterfactual treatments $\mathbf{T}$, $\mathbf{T}^{CF}$ as well as the counterfactual outcomes $\mathbf{A}^{CF}$. Then, the **second step** trains the graph learning model on both the observed factual data and created counterfactual data with the integrated loss function (Eq. (11)). Note that the discrepancy loss (Eq. (10)) is computed on the representations of node pairs learned by the graph encoder $f$, so the decoder $g$ is trained with data from both $\hat{P}^F$ and $\hat{P}^{CF}$ without balancing the constraints. Therefore, after the model is sufficiently trained, we freeze the graph encoder $f$ and fine-tune $g$ with only the factual data. Finally, after the decoder is sufficiently fine-tuned, we output the link prediction logits for both the factual and counterfactual adjacency matrices.

---

**Algorithm 1:** CFLP: Counterfactual graph learning for link prediction

**Input** : $f$, $g$, $\mathbf{A}$, $\mathbf{X}$, $n\_epochs$, $n\_epoch\_ft$
1 Compute $\mathbf{T}$ by Eq. (1) ;
2 Compute $\mathbf{T}^{CF}$, $\mathbf{A}^{CF}$ by Eqs. (3) and (4) ;
   /* model training                              */
3 Initialize $\Theta_f$ in $f$ and $\Theta_g$ in $g$ ;
4 **for** *epoch in range(n_epochs)* **do**
5 $\quad$ $\mathbf{Z} = f(\mathbf{A}, \mathbf{X})$ ;
6 $\quad$ Get $\widehat{\mathbf{A}}$ and $\widehat{\mathbf{A}}^{CF}$ via $g$ with Eqs. (6) and (7) ;
7 $\quad$ Update $\Theta_f$ and $\Theta_g$ with $\mathcal{L}$ ;       // (11)
8 **end**
   /* decoder fine-tuning                        */
9 Freeze $\Theta_f$ and re-initialize $\Theta_g$ ;
10 $\mathbf{Z} = f(\mathbf{A}, \mathbf{X})$ ;
11 **for** *epoch in range(n_epochs_ft)* **do**
12 $\quad$ Get $\widehat{\mathbf{A}}$ via $g$ with Eq. (6) ;
13 $\quad$ Update $\Theta_g$ with $\mathcal{L}_F$ ;         // Eq. (8)
14 **end**
   /* model inferencing                          */
15 $\mathbf{Z} = f(\mathbf{A}, \mathbf{X})$ ;
16 Get $\widehat{\mathbf{A}}$ and $\widehat{\mathbf{A}}^{CF}$ via $g$ with Eqs. (6) and (7) ;
**Output :** $\widehat{\mathbf{A}}$ for link prediction, $\widehat{\mathbf{A}}^{CF}$

---

**Complexity**  The complexity of the first step (finding counterfactual links with nearest neighbors) is proportional to the number of node pairs. When $\gamma$ is set as a small value to obtain indeed similar node pairs, this step (Eq. (3)) uses constant time. Moreover, the computation in Eq. (3) can be parallelized. Therefore, the time complexity is $O(N^2/C)$ where $C$ is the number of processes. For the complexity of the second step (training counterfactual learning model), the GNN encoder has time complexity of $O(LH^2N + LH|\mathcal{E}|)$ (Wu et al., 2020), where $L$ is the number of GNN layers and $H$ is the size of node representations. Given that we sample the same number of non-existing links as that of observed links during training, the complexity of a *three-layer MLP decoder* is $O(((H+1) \cdot d_h + d_h \cdot 1)|\mathcal{E}|) = O(d_h(H+2)|\mathcal{E}|)$, where $d_h$ is the number of neurons in the hidden layer. Therefore, the second step has linear time complexity w.r.t. the sum of node and edge counts.

**Limitations**  First, as mentioned above, the computation of finding counterfactual links has a worst-case complexity of $O(N^2)$. Second, CFLP performs counterfactual prediction with only a single treatment; however, there are quite a few kinds of graph structural information that can be considered as treatments. Future work can leverage the rich structural information by bundled treatments (Zou et al., 2020) in counterfactual graph learning.

# 4 Experiments

## 4.1 Experimental Setup

We conduct experiments on five benchmark datasets including citation networks (CORA, CITESEER, PUBMED (Yang et al., 2016)), social network (FACEBOOK (McAuley and Leskovec, 2012)), and drug-drug interaction network (OGB-DDI (Wishart et al., 2018)) from the Open Graph Benchmark

Table 2: Link prediction performances measured by Hits@20. Best performance and best baseline performance are marked with bold and underline, respectively.

| | CORA | CITESEER | PUBMED | FACEBOOK | OGB-DDI |
|---|---|---|---|---|---|
| Node2Vec | 49.96±2.51 | 47.78±1.72 | 39.19±1.02 | 24.24±3.02 | 23.26±2.09 |
| MVGRL | 19.53±2.64 | 14.07±0.79 | 14.19±0.85 | 14.43±0.33 | 10.02±1.01 |
| VGAE | 45.91±3.38 | 44.04±4.86 | 23.73±1.61 | 37.01±0.63 | 11.71±1.96 |
| SEAL | 51.35±2.26 | 40.90±3.68 | 28.45±3.81 | 40.89±5.70 | 30.56±3.86 |
| LGLP | 62.98±0.56 | 57.43±3.71 | – | 37.86±2.13 | – |
| GCN | 49.06±1.72 | 55.56±1.32 | 21.84±3.87 | 53.89±2.14 | 37.07±5.07 |
| GSAGE | 53.54±2.96 | 53.67±2.94 | 39.13±4.41 | 45.51±3.22 | 53.90±4.74 |
| JKNet | 48.21±3.86 | 55.60±2.17 | 25.64±4.11 | 52.25±1.48 | 60.56±8.69 |
| Our proposed CFLP with different graph encoders | | | | | |
| CFLP w/ GCN | 60.34±2.33 | 59.45±2.30 | 34.12±2.72 | 53.95±2.29 | 52.51±1.09 |
| CFLP w/ GSAGE | 57.33±1.73 | 53.05±2.07 | 43.07±2.36 | 47.28±3.00 | 75.49±4.33 |
| CFLP w/ JKNet | **65.57**±1.05 | **68.09**±1.49 | **44.90**±2.00 | **55.22**±1.29 | **86.08**±1.98 |

(OGB) (Hu et al., 2020). For the first four datasets, we randomly select 10%/20% of the links and the same numbers of disconnected node pairs as validation/test samples. The links in the validation and test sets are masked off from the training graph. For OGB-DDI, we used the OGB official train/validation/test splits. Statistics for the datasets are given in Table 1 and details are in Appendix. We use K-core (Bader and Hogue, 2003) clusters as the default treatment variable. We evaluate CFLP on three commonly used GNN encoders: GCN (Kipf and Welling, 2016a), GSAGE (Hamilton et al., 2017), and JKNet (Xu et al., 2018). We compare the link prediction performance of CFLP against Node2Vec (Grover and Leskovec, 2016), MVGRL (Hassani and Khasahmadi, 2020), VGAE (Kipf and Welling, 2016b), SEAL (Zhang and Chen, 2018), LGLP (Cai et al., 2021), and GNNs with MLP decoder. We report averaged test performance and their standard deviation over 20 runs with different random parameter initializations. Other than the most commonly used of Area Under ROC Curve (AUC), we report Hits@20 (one of the primary metrics on OGB leaderboard) as a more challenging metric, as it expects models to rank positive edges higher than nearly all negative edges.

Besides performance comparison on link prediction, we will answer two questions to suggest a way of choosing a treatment variable for creating counterfactual links: (Q1) Does CFLP sufficiently learn the observed *averaged treatment effect* (ATE) derived from the counterfactual links? (Q2) What is the relationship between the estimated ATE learned in the method and the prediction performance? If the answer to Q1 is yes, then the answer to Q2 will indicate how to choose treatment based on observed ATE. To answer the Q1, we calculate the observed ATE ($\widehat{\text{ATE}}_{obs}$) by comparing the observed links in $\mathbf{A}$ and created counterfactual links $\mathbf{A}^{CF}$ that have opposite treatments. And we calculate the estimated ATE ($\widehat{\text{ATE}}_{est}$) by comparing the predicted links in $\widehat{\mathbf{A}}$ and predicted counterfactual links $\widehat{\mathbf{A}}^{CF}$. Formally, $\widehat{\text{ATE}}_{obs}$ and $\widehat{\text{ATE}}_{est}$ are defined as

$$\widehat{\text{ATE}}_{obs} = \frac{1}{N^2} \sum_{i=1}^{N} \sum_{j=1}^{N} \{\mathbf{T} \odot (\mathbf{A} - \mathbf{A}^{CF}) + (\mathbf{1}_{N \times N} - \mathbf{T}) \odot (\mathbf{A}^{CF} - \mathbf{A})\}_{i,j}. \quad (12)$$

$$\widehat{\text{ATE}}_{est} = \frac{1}{N^2} \sum_{i=1}^{N} \sum_{j=1}^{N} \{\mathbf{T} \odot (\widehat{\mathbf{A}} - \widehat{\mathbf{A}}^{CF}) + (\mathbf{1}_{N \times N} - \mathbf{T}) \odot (\widehat{\mathbf{A}}^{CF} - \widehat{\mathbf{A}})\}_{i,j}. \quad (13)$$

The treatment variables we will investigate are usually graph clustering or community detection methods, such as K-core (Bader and Hogue, 2003), stochastic block model (SBM) (Karrer and Newman, 2011), spectral clustering (SpecC) (Ng et al., 2001), propagation clustering (PropC) (Raghavan et al., 2007), Louvain (Blondel et al., 2008), common neighbors (CommN), Katz index, and hierarchical clustering (Ward) (Ward Jr, 1963). We use JKNet (Xu et al., 2018) as the default graph encoder.

Implementation details and supplementary experimental results (e.g., sensitivity on $\gamma$, ablation study on $\mathcal{L}_{CF}$ and $\mathcal{L}_{disc}$) can be found in Appendix. Source code is available in supplementary material.

## 4.2 Experimental Results

**Link Prediction** Tables 2 and 3 show the link prediction performance of Hits@20 and AUC by all methods. LGLP on PUBMED and OGB-DDI are missing due to the out of memory error when

Table 3: Link prediction performances measured by AUC. Best performance and best baseline performance are marked with bold and underline, respectively.

|  | CORA | CITESEER | PUBMED | FACEBOOK | OGB-DDI |
|---|---|---|---|---|---|
| Node2Vec | 84.49±0.49 | 80.00±0.68 | 80.32±0.29 | 86.49±4.32 | 90.83±0.02 |
| MVGRL | 75.07±3.63 | 61.20±0.55 | 80.78±1.28 | 79.83±0.30 | 81.45±0.99 |
| VGAE | 88.68±0.40 | 85.35±0.60 | 95.80±0.13 | 98.66±0.04 | 93.08±0.15 |
| SEAL | 92.55±0.50 | 85.82±0.44 | 96.36±0.28 | **99.60**±0.02 | 97.85±0.17 |
| LGLP | 91.30±0.05 | 89.41±0.13 | – | 98.51±0.01 | – |
| GCN | 90.25±0.53 | 71.47±1.40 | 96.33±0.80 | 99.43±0.02 | 99.82±0.05 |
| GSAGE | 90.24±0.34 | 87.38±1.39 | 96.78±0.11 | 99.29±0.04 | 99.93±0.02 |
| JKNet | 89.05±0.67 | 88.58±1.78 | 96.58±0.23 | 99.43±0.02 | **99.94**±0.01 |
| Our proposed CFLP with different graph encoders |  |  |  |  |  |
| CFLP w/ GCN | 92.55±0.50 | 89.65±0.20 | 96.99±0.08 | 99.38±0.01 | 99.44±0.05 |
| CFLP w/ GSAGE | 92.61±0.52 | 91.84±0.20 | 97.01±0.01 | 99.34±0.10 | 99.83±0.05 |
| CFLP w/ JKNet | **93.05**±0.24 | **92.12**±0.47 | **97.53**±0.17 | 99.31±0.04 | **99.94**±0.01 |

running the code package from the authors. We observe that our CFLP on different graph encoders achieve similar or better performances compared with baselines. The only exception is the AUC on FACEBOOK where most methods have close-to-perfect AUC. As AUC is a relatively easier metric comparing with Hits@20, most methods achieved good performance on AUC. We observe that CFLP with JKNet almost consistently achieves the best performance and outperforms baselines significantly on Hits@20. Specifically, compared with the best baseline, CFLP improves relatively by 16.4% and 0.8% on Hits@20 and AUC, respectively. It is worth noting that CFLP with JKNet achieves the state-of-the-art performance on the official leaderboard[1] of OGB-DDI.

Figure 2 shows the AUC performance of CFLP on CORA with different combinations of $\alpha$ and $\beta$. We observe that the performance is the poorest when $\alpha = \beta = 0$ and gradually improves and gets stable as $\alpha$ and $\beta$ increase, showing that CFLP is robust to the hyperparameters $\alpha$ and $\beta$.

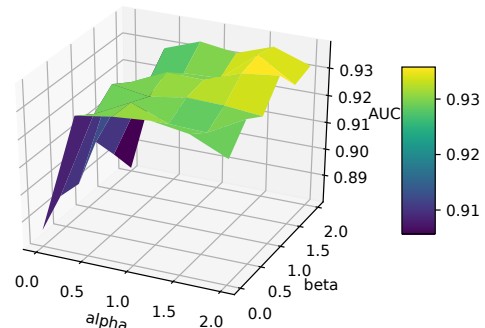

Figure 2: AUC performance of CFLP on CORA w.r.t different combanitions of $\alpha$ and $\beta$.

**ATE with Different Treatments** Tables 4 and 5 show the link prediction performance, $\widehat{\text{ATE}}_{obs}$, and $\widehat{\text{ATE}}_{est}$ of CFLP (with JKNet) when using different treatments. The treatments in Tables 4 and 5 are sorted by the Hits@20 performance. Bigger ATE indicates stronger causal relationship between the treatment and outcome, and vice versa. We observe: (1) $\widehat{\text{ATE}}_{est}$ values are generally close to $\widehat{\text{ATE}}_{obs}$, showing that CFLP was sufficiently trained to learn the causal relationship between graph structure information and link existence; (2) $\widehat{\text{ATE}}_{obs}$ and $\widehat{\text{ATE}}_{est}$ are both negatively correlated with the link prediction performance, showing that we can pick a proper treatment prior to training a model with CFLP. Using the treatment that has the weakest causal relationship with link existence is likely to train the model to capture more essential factors on the outcome, in a way similar to denoising the unrelated information from the representations.

## 5 Related Work

**Link Prediction** With its wide applications, link prediction has draw attention from many research communities including statistical machine learning and data mining. Stochastic generative methods based on stochastic block models (SBM) are developed to generate links (Mehta et al., 2019). In data mining, matrix factorization (Menon and Elkan, 2011), heuristic methods (Philip et al., 2010; Martínez et al., 2016), and graph embedding methods (Cui et al., 2018) have been applied to predict links in the graph. Heuristic methods compute the similarity score of nodes based on their neighborhoods. These

---

[1]https://ogb.stanford.edu/docs/leader_linkprop/#ogbl-ddi

Table 4: Results of CFLP with different treatments on CORA. (sorted by Hits@20)

| | Hits@20 | $\widehat{\text{ATE}}_{obs}$ | $\widehat{\text{ATE}}_{est}$ |
|---|---|---|---|
| K-core | 65.6±1.1 | 0.002 | 0.013±0.003 |
| SBM | 64.2±1.1 | 0.006 | 0.023±0.015 |
| CommN | 62.3±1.6 | 0.007 | 0.053±0.021 |
| PropC | 61.7±1.4 | 0.037 | 0.059±0.065 |
| Ward | 61.2±2.3 | 0.001 | 0.033±0.012 |
| SpecC | 59.3±2.8 | 0.002 | 0.033±0.011 |
| Louvain | 57.6±1.8 | 0.025 | 0.138±0.091 |
| Katz | 56.6±3.4 | 0.740 | 0.802±0.041 |

Table 5: Results of CFLP with different treatments on CITESEER. (sorted by Hits@20)

| | Hits@20 | $\widehat{\text{ATE}}_{obs}$ | $\widehat{\text{ATE}}_{est}$ |
|---|---|---|---|
| SBM | 71.6 ±1.9 | 0.004 | 0.005 ±0.001 |
| K-core | 68.1±1.5 | 0.002 | 0.010±0.002 |
| Ward | 67.0±1.7 | 0.003 | 0.037±0.009 |
| PropC | 64.6±3.6 | 0.141 | 0.232±0.113 |
| Louvain | 63.3±2.5 | 0.126 | 0.151±0.078 |
| SpecC | 59.9±1.3 | 0.009 | 0.166±0.034 |
| Katz | 57.3±0.5 | 0.245 | 0.224±0.037 |
| CommN | 56.8±4.9 | 0.678 | 0.195±0.034 |

methods can be generally categorized into first-order, second-order, and high-order heuristics based on the maximum distance of the neighbors. Graph embedding methods learn latent node features via embedding lookup and use them for link prediction (Perozzi et al., 2014; Tang et al., 2015; Grover and Leskovec, 2016; Wang et al., 2016).

In the past few years, GNNs have showed promising results on various graph-based tasks with their ability of learning from features and custom aggregations on structures, (Kipf and Welling, 2016a; Hamilton et al., 2017; Xu et al., 2018; Wu et al., 2020). With node pair representations and an attached MLP or inner-product decoder, GNNs can be used for link prediction (Zhang et al., 2020a). For example, VGAE used GCN to learn node representations and reconstruct the graph structure (Kipf and Welling, 2016b). SEAL extracted a local subgraph around each target node pair and then learned graph representation from local subgraph for link prediction (Zhang and Chen, 2018). Following the scheme of SEAL, Cai and Ji (2020) proposed to improve local subgraph representation learning by multi-scale graph representation learning. And LGLP inverted the local subgraphs to line graphs before learning representations (Cai et al., 2021). However, very limited work has studied to use causal inference for improving link prediction.

**Counterfactual Prediction**   As a mean of learning the causality between treatment and outcome, counterfactual prediction has been used for a variety of applicaitons such as recommender systems (Wang et al., 2020; Xu et al., 2020), health care (Alaa and van der Schaar, 2017), vision-language tasks (Zhang et al., 2020b; Parvaneh et al., 2020), and decision making (Coston et al., 2020; Pitis et al., 2020; Kusner et al., 2017). To infer the causal relationships, previous work usually estimated the ITE via function fitting models (Gelman and Hill, 2006; Chipman et al., 2010; Wager and Athey, 2018; Assaad et al., 2021) which estimated the transductive ITE. Peysakhovich et al. (2019) and Zou et al. (2020) studied counterfactual prediction with multiple agents and bundled treatments, respectively. Pawlowski et al. (2020) proposed a deep structural causal model for tractable counterfactual inference.

**Causal Inference**   Causal inference methods usually re-weighted samples based on propensity score (Rosenbaum and Rubin, 1983; Austin, 2011; Kuang et al., 2017a,b) to remove confounding bias from binary treatments. Recently, several works studied about learning treatment invariant representation to predict the counterfactual outcomes (Hassanpour and Greiner, 2019b,a; Shalit et al., 2017; Yao et al., 2018; Bica et al., 2020; Hassanpour and Greiner, 2019a; Li and Fu, 2017). When part of unobserved outcomes may mislead the counterfactual prediction, Louizos et al. (2017) attempted to infer the outcomes from proxies, and Hartford et al. (2017) introduced instrumental variable. SITE preserved local similarity to balance the distributions of control and treated groups (Yao et al., 2018). Yoon et al. (2018) estimated ITE with generative adversarial networks (GANs). Assaad et al. (2021) discussed the trade-off between achieving balance and predictive power.

# 6   Conclusion

In this work, we presented a counterfactual graph learning method for link prediction (CFLP). We introduced the idea of counterfactual prediction to improve link prediction on graphs. CFLP accurately predicted the missing links by exploring the causal relationship between global graph structure and link existence. Extensive experiments demonstrated that CFLP achieved the state-of-the-art performance on benchmark datasets.

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
