# OpenReview forum: "Counterfactual Graph Learning for Link Prediction"
_NeurIPS.cc/2021/Conference — NeurIPS 2021 Submitted_

### Official Review · Reviewer_eUXt · 2021-07-15

**Rating:** 5
**Confidence:** 3

**Summary:**

This paper introduces the idea of counterfactual prediction to improve link prediction on graphs and proposes a framework, counterfactual graph learning method for link prediction (CFLP). Specifically, the authors first introduce a method of generating counterfactual samples and then utilize these new data as well as original factual data to train the proposed model. The proposed method outperforms competitive baseline methods on several benchmark datasets in the link prediction task. The authors also provide sensitivity analysis and examine the impact of different treatments (i.e., results from different community detection algorithms) to link prediction results.

**Limitations And Societal Impact:**

The authors point out that the proposed method has limitations in terms of computation efficiency and the formulation of treatment. The authors do not provide potential negative societal impacts of this work. To this end, I suggest the authors consider the unfairness problem that might exist in this proposed method.

**Main Review:**

Originality:

The idea of introducing counterfactual learning into link prediction is novel. The authors introduce a new method of link prediction that combines graph neural networks and counterfactual learning.

Quality:

- The theoretical background is not clear enough. The formulated problem in this paper is to estimate individual treatment effects (ITE). In causal inference, assumptions are required in order to make individual causal effects identifiable, e.g., consistency, no-hidden confounders, etc. The authors should clarify and discuss these assumptions.

- The authors claim that employing counterfactual learning can help train the model to capture more essential factors on the outcome. They obtain results of ATE and link prediction based on different treatments (see Table 4 and 5). However, the results are a little confusing. It is unclear how this method captures essential factors. The authors are supposed to provide more results or explanations to support their claim.

Clarity:

+ Overall, this paper is well organized. The technical part is mostly clear.

- The problem proposed in the introduction is not fully supported by the experiments. Specifically, the authors claim that answering counterfactual questions can help capture essential factors to accurately predict missing links in the test data. There is no evidence to show how this works.

- The authors propose an approximation for finding counterfactual links (Eq. 3). There are no experimental results to show that the approximation method produces results similar to those in Eq. 2. Will different counterfactual link generation methods affect the model's link prediction results?

- The introduction of Balancing Counterfactual Learning is not very clear. The authors should introduce the formal definitions of PˆF_f and PˆCF_f in Eq. 10.

- In Figure 2, the results show that beta (balancing counterfactual learning) does not have much influence on AUC performance. Meanwhile, as the authors mentioned, AUC is an easier metric than Hits@20. The sensitivity results on Hits@20 are missing.
Eq. 8 and 9 are unclear. The training is conducted only on the training data. The loss should not be calculated based on all links, e.g., for loop N nodes.

- The link prediction results need more explanation (see Significance).

Significance:

- In this work, the authors first generate counterfactual samples and then train the proposed method based on the new data and original data. The contribution of this work seems limited except involving the concept of counterfactual learning. In short, the authors seem to add more samples to train a model for improving the link prediction performance.

- The link prediction results need more explanation. Why does counterfactual learning help link prediction? There is a jump from treatment effects learning to link prediction. Meanwhile, since baselines are trained on original data (without counterfactual data), is this comparison fair?

- There are some related works that should be included. The following published work seems to have achieved better AUC results in link prediction than the proposed method in this paper on the Cora and Citeseer datasets.

(1) Davidson, T.R., Falorsi, L., De Cao, N., Kipf, T. and Tomczak, J.M., 2018. Hyperspherical variational auto-encoders. arXiv preprint arXiv:1804.00891.

(2) Yang, H., Pan, S., Zhang, P., Chen, L., Lian, D. and Zhang, C., 2018, November. Binarized attributed network embedding. In 2018 IEEE International Conference on Data Mining (ICDM) (pp. 1476-1481). IEEE.

Minor issues:
- It would be better to describe terms consistently, e.g., individualized treatment effect and individual treatment effect.
- It would be clearer to draw balancing counterfactual learning in Fig.1(b).
- It would be helpful to explain the ratio of created CF samples for each dataset (i.e., how many samples satisfy Eq.3). This might help evaluate the performance of the proposed model.

Thanks to the authors for the response.



**Time Spent Reviewing:**

8

---

> ### Author Response · Authors · 2021-08-09
> **Our response**
>
> Dear reviewer eUXt, we appreciate your careful reading and insightful comments. Following are detailed responses regarding your concerns.
>
> 1. [Quality] Assumptions on causal inference.
>
> Our proposed model is designed based on the basic triangle structural causal model (three-nodes DAG). As we mentioned in the introduction and also introduced in Sec. 3.1, our causal DAG contains the following nodes and edges:
>
> W (Confounder; unobservable): latent node pair representations,\
> X (Treatment; observable): graph structural information (e.g., whether two nodes belong to the same community/cluster),\
> Y (Outcome; observable): link existence.\
> Edges: W $\rightarrow$ X, W $\rightarrow$ Y, and X $\rightarrow$ Y.
>
> The model follows standard assumptions for counterfactual representation learning [1]. We will add the causal DAG in our paper to improve the clarity. We want to point out that our main contribution is to use causal inference for improving graph machine learning; it does not aim at discovering novel causal factors. The formulated problem is not to find a new method to estimate ITE. It is to use the generated counterfactual data based on estimated ATE to improve the link prediction performance. We show that the estimated ATE of our defined treatment variables has negative correlation with the link prediction performance (Tables 4 and 5) and using the clusterings of the lowest estimated ATE as treatment in our proposed graph learning model can effectively improve the link prediction performance (Tables 2 and 3).
>
> [1]  “Learning Representations for Counterfactual Inference.” ICML 2016.
>
> 2. [Quality] How the proposed CFLP captures essential factors.
>
> As argued in lines 20-34 (page 1), conventional link prediction models would be confused by non-essential factors such as global structural roles (e.g., the nodes belonging to the same community). We invent a model that creates additional training examples to learn against the non-essential factors so that the model will capture more essential factors related to link prediction. In detail, as described in lines 59-66 (page 2), our model first generates the counterfactual samples with the opposite global structural roles and then learns from both factual and counterfactual data. Tables 4 and 5 and lines 304-308 (page 8) provided experimental results and analysis to support our claim: the treatments defined by global structural roles are non-essential factors (small ATE values) for link prediction and the model captures more essential factors (though unobservable) by excluding these non-essential factors.
>
> 3. [Clarity] Effect of approximation for Eq. (3) w.r.t. Eq. (2).
>
> The ideal distance measurement $d((v_a,v_b), (v_i, v_j))$ in Eq.(2) that generates the best counterfactual examples for link prediction is missing because the idea of incorporating counterfactual links into graph learning models is new (and proposed in this work). Eq. (3) assumes that the sum of distances between the nodes is an effective approximation of the distance between node pairs and experimental results in Tables 2 and 3 show its effectiveness. We hypothesize that this measurement (Eq. (3)) from the ideal distance measurement (Eq. (2)). However, we are not able to find a good theoretical foundation to precisely identify the ideal distance measurement as the idea is newly proposed in this work.
>
> 4. [Clarity] Definition of of $\hat{P}^F_f$ and $\hat{P}^{CF}_f$.
>
> Thanks for pointing this out. $\hat{P}^F$ and $\hat{P}^{CF}$ are defined in lines 155-157 (page 4), which are the empirical factual and counterfactual distributions, respectively. $\hat{P}^F_f$ and $\hat{P}^{CF}\_f$ are their representations learned by $f$, i.e., the representation for $(v_i, v_j, T\_{i,j}^F)$ is $[\mathbf{z}\_i \odot \mathbf{z}\_j, T\_{i,j}]$ (Eq. (6)) and the representation for $(v_i, v_j, T\_{i,j}^{CF})$ is $[\mathbf{z}\_i \odot \mathbf{z}\_j, T\_{i,j}^{CF}]$ (Eq. (7)). We will add this after Eq. (10) to improve the clarity.
>
> 5. [Clarity] Sensitivity results on Hits@20.
>
> We performed sensitivity analysis on Hits@20 and found it similar to the AUC one (Fig. 2). So we didn’t include it due to space limitations. We will add the figure of alpha and beta vs Hits@20.
>
> 6. [Clarity] Clarification on Eq. (8) and (9).
>
> As mentioned in line 80-81 (page 3), $\mathbf{A}$ denotes the adjacency matrix for the observed graph, which does not include the validation and test data as those links were masked off (as a standard experimental setting for link prediction [2-4]). Hence looping through all nodes calculates the loss only on training data (same as [5]). We will add such explanations after Eq. (9) to improve the clarity.
>
> [2] “Link Prediction Based on Graph Neural Networks.” NeurIPS 2018.\
> [3] “Line graph neural networks for link prediction.” TPAMI 2021.\
> [4] “Open Graph Benchmark: Datasets for Machine Learning on Graphs.” NeurIPS 2020.\
> [5] “Variational Graph Auto-Encoders.” Bayesian Deep Learning Workshop at NeurIPS 2016.
>
> 7. [Significance] Why counterfactual learning helps link prediction and is using counterfactual data fair.
>
> As answered in previous questions, counterfactual learning helps link prediction because the use of counterfactual data helps the model to capture more essential information by excluding non-essential information. Moreover, counterfactual data was generated from original data by our method without any new required input or information. Defining and generating the counterfactual links is one of the main contributions of this work. Therefore, the comparison is fair because our method generates the counterfactual data and does not require any additional input.
>
> 8. [Significance] Missing related works.
>
> We will cite, include, and discuss these papers in related work. To clarify the performance, the numbers reported in these two papers are not directly comparable with ours because they had different experimental settings. We used 70%/10%/20% of the edges for the training/validation/testing sets, while [6] had a split of 85%/5%/10% and [7] used 90% edges for training and the rest for testing. Their use of more training data resulted in better AUC scores. We choose very competitive models as baselines in our experiments. For example: SEAL [8] is currently ranked as the first or second places in multiple link prediction benchmarks on Open Graph Benchmark (OGB) leaderboard; LGLP [9] outperformed SEAL on many traditional link prediction benchmarks. For example, on the Citeseer dataset, as shown in Table 3, our proposed CFLP outperformed SEAL and CFLP relatively by 7.3% and 3.0% on AUC score, respectively.
>
> [6] “Hyperspherical Variational Auto-encoders.” UAI 2018.\
> [7] “Binarized Attributed Network Embedding.” ICDM 2018.\
> [8] “Link prediction based on graph neural networks.” NeurIPS 2018.\
> [9] “Line graph neural networks for link prediction.” TPAMI 2021.
>
> 9. Minor issues.
>
> Thank you for pointing these out, we will update according to your suggestions.
>
> 10. [Societal impact] Consideration of the unfairness problem.
>
> Thanks for your suggestion. The potential fairness issue in decisions of graph machine learning methods are indeed important. For example, when the method is used in applications that have different demographic groups, the developers should consider the fairness-specific design [10,11] to ensure fairness in deployment. We will add discussion of the fairness issue in the paper.
>
> [10] “Learning Certified Individually Fair Representations.” NeurIPS 2020.\
> [11] “Ensuring Fairness Beyond the Training Data.” NeurIPS 2020.

---

### Official Review · Reviewer_oDMj · 2021-07-16

**Rating:** 6
**Confidence:** 4

**Summary:**

In this paper, the authors consider a link prediction problem utilizing the idea of causal inference.
To find the essential factors between nodes, they make counterfactual links using a matching-like algorithm for pairs of nodes. A GNN-based model is trained to learn representations so that the model can predict both the factual and counterfactual links. In training, first they jointly optimize the entire loss function including the counterfactual loss and the balancing loss, and then they fix the encoder and only train the decoder using the factual data. In the link prediction experiment, they use public popular datasets to validate the proposed method and evaluate in terms of some metrics such as AUC and HitRatio. They also analyze the proposal in terms of ATE and discover negative correlation between the performance and ATE.
The main contributions of this paper are as follows: 1) This is the first work that introduce the idea of causal inference into link prediction task. 2) They make counterfactual links and trained GNN-based methods to learn the causal relationship. 3) The proposed method outperforms the strong baselines.

**Limitations And Societal Impact:**

They approximate Eq. 2 to make counterfactual links for the sake of fast computation, the effect of approximation should be provided because it requires the additional hyper-parameter gamma.
It would be better to provide theoretical analysis.

**Main Review:**

This paper is well written and easy to follow.
I find the idea to incorporate causal inference into link prediction is novel and interesting. I thought the hyper-parameter gamma, that defines the maximum distance of two nodes are similar is quite important value because that defines the counterfactual links but in the supplement, they showed CFLP is robust to the choice of gamma. That is one of the benefits of CFLP and it would be better to provide some analysis why CFLP is robust to gamma.
In the experiment, the authors explain that they sample the same number of disconnected pairs as the links when validation. However, this should be improved because the ratio of positive and negative links is totally balanced and does not match that of the entire dataset. It also would be better to evaluate the results in terms of Precision and Recall.  Though the evaluation can be improved, the experimental results are convincing.

**Time Spent Reviewing:**

5 hours

---

> ### Author Response · Authors · 2021-08-09
> **Our response**
>
> Dear reviewer oDMj, we appreciate your careful reading and insightful comments. Following are detailed responses regarding your concerns.
>
> 1. Analysis on why CFLP is robust to $\gamma$.
>
> As shown in Figure 3(a-d) and lines 628-630 (Page 16), CFLP achieves generally good results when $\gamma_{pct}$ is between 10% and 20%. To clarify at this point, we do not claim that CFLP is robust to any choice of $\gamma$. For example, as shown in Figure 3(c) and (d), when $\gamma$ is too high ($\gamma_{pct} \ge 25$%), the Hits@20 drops significantly. This is because a too high $\gamma$ introduces too much noise -- the selected counterfactual link $(v_a, v_b)$ is not similar enough to the target link $(v_i, v_j)$. On the other hand, we performed data analysis to explain why CFLP’s performance looks good when $\gamma_{pct}$ is in the range of 10-20%. The values of the distance variable $d(x_i, x_a)+d(x_j, x_b)$ follow a long-tail distribution (value vs. rank, sorted from smallest distance to biggest). The head of the distribution often stops at a “relative” rank position between 10% and 20%. So the most similar counterfactual links (at the top 10%) would have been included when $\gamma_{pct}$ is between 10-20%. And more links that were ranked after 20% would have too high a distance to be helpful being used as counterfactual examples.
>
> 2. About evaluation settings.
>
> Regarding the evaluation settings for link prediction, researchers are commonly using balanced (positive and negative) links and AUC score to set up a prediction task to evaluate the models. This is because of two reasons: (1) as the models did not see any links in the test set, no matter positive or negative links, it is fair to say a higher AUC indicates a better prediction performance; (2) using balanced test samples to evaluate can make the evaluation scores explainable, for example, random guess would achieve an AUC score of 0.5. If we artificially made the test data to be imbalanced at a specific ratio or directly use the ratio of the original imbalance dataset, we would lose the above two advantages: The evaluation results might no longer be generalizable or explainable. Due to these reasons, most existing works have used balanced sampling to evaluate link prediction models such as [1-5]. Following these works, we also employ the popular Hits@K (K = 20 or 50) and Average Precision (AP) measures to evaluate the models.
>
> [1] “Link Prediction Based on Graph Neural Networks.” NeurIPS 2018.\
> [2] “Line graph neural networks for link prediction.” TPAMI 2021.\
> [3] “Hyperbolic Graph Convolutional Neural Networks.” NeurIPS 2019.\
> [4] “Variational Graph Auto-Encoders.” Bayesian Deep Learning Workshop at NeurIPS 2016.\
> [5] “Open Graph Benchmark: Datasets for Machine Learning on Graphs.” NeurIPS 2020.
>
> 3. Effect of approximation for Eq. (3) w.r.t. Eq. (2).
>
> Thanks for this suggestion. We are interested in theoretical analysis on the effect of approximation as well as the effect of hyper-parameter $\gamma$. However, the ideal distance measurement $d((v_a,v_b), (v_i, v_j))$ in Eq.(2) that generates the best counterfactual examples for link prediction is missing. Once the ideal measurement could be found, we would be able to use the approximated measurement (in Eq.(3)) to analyze the theoretical bound between their effects. Unfortunately, we are not able to find a good theoretical foundation to precisely identify the ideal distance measurement as the idea is newly proposed in this work. We will explore the theory that has potential to derive the ideal distance measurement in the future work.

---

### Official Review · Reviewer_fY1x · 2021-07-16

**Rating:** 3
**Confidence:** 4

**Summary:**

The paper consider the task of counterfactually predicting edges in attributed graphs. I think this is a good effort (but incomplete) in a consequential and fast growing area. The paper needs to define the counterfactual task formally and prove their approach really does what the paper claim it does.

**Ethical Concerns:**

No ethical concerns

**Limitations And Societal Impact:**

No, the authors did not adequately addressed the limitations. The work talks about counterfactual link prediction but does not provide guarantees that its counterfactual estimator is correct.

**Main Review:**

My main concern with this work is the absence of a clear counterfactual model and a proof that the approach indeed does counterfactual inference according to the model. I am unable to evaluate the correctness of the approach without a precise theoretical definition of the task and a proof that the method indeed can produce a counterfactual query.

+ Very important task. I am excited to see work in this space. I encourage the authors to continue improving their paper.

- No structural causal model or even a causal DAG. This alone is grounds for rejection, since the reviewers cannot evaluate the correctness of the approach.

-In graphs, variables are often dependent. What is the causal mechanism that allows us to say that the "counterfactual link" (va,vb) is really a natural experiment of (vi,vj)? That is, please describe in mathematical terms how clustering affects the structural causal model and mathematically prove it allows counterfactual inference. Then, please prove that the proposed form of counterfactual inference is the correct. Ideally, by showing error bounds.

- In "3.1: Treatment": It is unclear how the treatment is related to the hidden and observed variables in the task. How are the clusters described in the structural causal model?

- When talking about "Counterfactual Links", the paper still offers no structural causal model.

- The missing related work (Bevilacqua et al. 2021) tackles graph classification under a counterfactually-driven data shift. That work shows the structural causal model and proves a bound that the representation is robust to the counterfactual shifts.

- The embedding of (Hassani and Khasahmadi, 2020) is a structural node embedding used to predict node attributes or graph labels (not links). It makes sense to use a structural embedding for d(xi,xa) but the use of GCN with H_0 = X (after Equation (5)) is puzzling... with H_0 = X rather than the GCN one-hot encoding (used for link prediction in Kipf and Welling) gives structural node embeddings, which is not to be used for link prediction.

Bevilacqua, Beatrice, Yangze Zhou, and Bruno Ribeiro. "Size-Invariant Graph Representations for Graph Classification Extrapolations." ICML (2021)

**Time Spent Reviewing:**

2

---

> ### Author Response · Authors · 2021-08-09
> **Our response**
>
> Dear reviewer fY1x, we appreciate your careful reading and insightful comments. Following are the detailed responses regarding your concerns.
>
> 1. About the missing DAG of causal model and the position of clusters in the structural causal model.
>
> We want to point out that the main contribution of this work is not a novel structural causal model for causal analysis. Our main contribution is that we take advantage of existing basic yet effective causal models to enhance graph machine learning methods for link prediction. We did not include the causal DAG behind our proposed model as it is the very basic three-nodes DAG. As we mentioned in the introduction and also introduced in Sec. 3.1, our causal DAG contains the following nodes and edges:
>
> W (Confounder; unobservable): latent node pair representations,\
> X (Treatment; observable): graph structural information (e.g., whether two nodes belong to the same community/cluster),\
> Y (Outcome; observable): link existence.\
> Edges: W $\rightarrow$ X, W $\rightarrow$ Y, and X $\rightarrow$ Y.
>
> We will add the causal DAG in our paper to improve the clarity. However, we argue that different from related work on causal inference that aims to estimate or predict ATE(X), our main contribution is to use the estimated ATE(X) to first pick treatment, then create counterfactual links, and ultimately improve the link prediction performance.
>
> As argued in lines 20-34 (page 1), conventional link prediction models would be confused by non-essential factors such as global structural roles (e.g., belonging to the same community). We employ a basic causal model in which the graph structural factors are defined as treatment.  Then link prediction models learn node pair representations from both observed links and counterfactual links that are created to exclude the non-essential information for better understanding link existence. Tables 4 and 5 and lines 304-308 (page 8) provided experimental results and analysis to support our claim: the treatments defined by global structural roles are non-essential factors (small ATE values) for link prediction and the model has captured more essential factors by excluding these non-essential factors. Finally, we want to emphasize the interdisciplinary impact -- this work sheds insights that a good use of causal models (even basic ones) can greatly improve the performance of (graph) machine learning tasks, which in our case is achieving state-of-the-art performance on link prediction.
>
> 2. Missing related work.
>
> Thank you for pointing this out, we will include the discussion on this work in Section 5.
>
> 3. About the use of GCN with $\mathbf{H}^{(0)} = \mathbf{X}$.
>
> We argue that using raw node features as the input features ($\mathbf{H}^{(0)}$) is the standard approach in GNN literature [1-3] (also used in GCN [1]) and one-hot encoding is usually used when the graph data do not have any raw node features.
>
> [1] “Semi-supervised Classification with Graph Convolutional Networks.” ICLR 2017.
>
> [2] “Variational Graph Auto-Encoders.” Bayesian Deep Learning Workshop at NeurIPS 2016.
>
> [3] “Inductive Representation Learning on Large Graphs.” NeurIPS 2017.

---

> > ### Comment · Reviewer_fY1x · 2021-08-27
> > **Incorrect DAG**
> >
> > I think it is reasonable to assume your graph contains more than two nodes. Hence, X cannot be just a pair of nodes. The DAG must contain all nodes and edges. The entire graph is one single observation.

---

> > > ### Author Response · Authors · 2021-08-27
> > > **Response to Reviewer fY1x**
> > >
> > > Dear reviewer fY1x, thanks for your comments. In the task of link prediction, algorithms take each node pair as one single observation instead of taking the whole graph (dataset) as one observation. For example, SEAL [1] used a node pair as a data object and extracted its features from the local subgraph that contains the target node pair. Then it learned a mapping from each local subgraph (feature space) to link existence (label space).
> > >
> > > To complement with the above clarification, we’d like to offer “sample data tables” to present how the DAG and its variables were defined in our study. Suppose we have “ice cream sales” data as below, which has been widely used in traditional causal studies and even lectures that introduce the DAG of basic causal models:
> > >
> > > |         | Hidden factor (W): Weather (temperature)| Treatment (X): Sunburn | Outcome (Y): Ice cream sales |
> > > |:-------:|:---------------------------------------:|:----------------------:|:----------------------------:|
> > > | March 1 |                   96°F                  |           Yes          |              Yes             |
> > > | March 2 |                   83°F                  |           No           |              No              |
> > > | March 3 |                   93°F                  |           No           |              Yes             |
> > > | March 4 |                   85°F                  |           Yes          |              No              |
> > > |   ...   |                                         |                        |                              |
> > >
> > > Here comes an analog of the basic DAG ($W \rightarrow X$, $W \rightarrow Y$, and $X \rightarrow Y$). In our study (graph learning for link prediction), given a graph $G(\mathcal{V}, \mathcal{E})$ with nodes $\mathcal{V} = \\{v_1, v_2, v_3, ...\\}$, our data table looks like this:
> > >
> > > |       | Hidden factor (W): Latent node pair representations | Treatment (X): Graph structural information | Outcome (Y): Link existence |
> > > |:-----:|:---------------------------------------------------:|:-------------------------------------------:|:---------------------------:|
> > > | $v_1, v_2$ |   [0.93, ...] ($\mathbf{z}_1 \odot \mathbf{z}_2$)   | Yes (i.e., $v_1$ and $v_2$ in same cluster) |             Yes             |
> > > | $v_1, v_3$ |                     [0.83, ...]                     |                      No                     |              No             |
> > > | $v_1, v_4$ |                     [0.93, ...]                     |                      No                     |             Yes             |
> > > | ...        |                                                     |                                             |                             |
> > > | $v_2, v_3$ |                     [0.85, ...]                     |                     Yes                     |              No             |
> > > | ...        |                                                     |                                             |                             |
> > >
> > > **Rows are data objects (e.g., days, node pairs). Columns are variables.**
> > >
> > > Hope this response helps understand our study. If you have any questions, please feel free to let us know. Thanks again for your time and help.
> > >
> > > [1] “Link Prediction Based on Graph Neural Networks.” NeurIPS 2018.

---

> > > > ### Comment · Reviewer_fY1x · 2021-08-27
> > > > **Graphical models and graphs**
> > > >
> > > > The graphical model describing a graph can be broken down into independent pairs of node only through some assumptions. In observational tasks one does not need to describe the modeling assumptions in detail. But when claiming robustness to a counterfactual change, unfortunately, the precise mathematical definition of the model is necessary. Causal models on graphs are significantly more complex than causal models on iid data.
> > > >
> > > > For instance, assuming infinite exchangeability, one could use the Aldous-Hoover theorem to describe a mixture model where edges are conditionally independent given the mixture sample. This, however, requires describing a hierarchical model. Even this hierarchical model, however, makes strong assumptions. Infinite exchangeability is different from finite exchangeability (Persi Diaconis has an excellent paper on finite exchangeability (Finite forms of de Finetti's theorem on exchangeability)). Under finite exchangeability your model would need to be quite different. ​
> > > >
> > > > It could be that the authors are assuming a graph model akin to a stochastic block model (but even that is not 100% clear). Maybe  it is a factor model (David Blei's group has been looking at factor models for counterfactual inference (blessing of multiple causes)). If that is the case, I have another host of other questions to ask connected to your task. However, this is engaging in hypotheticals. I would like to see a precise mathematical statement of the SCM before I can judge whether the proposed method is sound.

---

> > > > > ### Comment · Reviewer_fY1x · 2021-08-27
> > > > > **Maybe this will help frame the discussion**
> > > > >
> > > > > https://www.stat.cmu.edu/~cshalizi/networks/16-2/lectures/02/malik.pdf
> > > > >
> > > > > See page 5 for a hierarchical factor model of under the infinite exchangeability assumption (proof of the equivalence between infinite exchangeability and this graphical model summarized in page 6...the actual proof can be found in Aldous 1983 paper https://www.stat.berkeley.edu/~aldous/Papers/me22.pdf)

---

> > > > > ### Author Response · Authors · 2021-08-28
> > > > > **Response to Reviewer fY1x**
> > > > >
> > > > > We appreciate Reviewer fY1x for pointing us to the literature and lecture material of the research field of causal modeling. It took us several hours to digest the papers, lecture notes, and online resources. We are happy to be aware of the existence of “significantly complex” causal models.
> > > > >
> > > > > However, as we have claimed in our submission and previous response, the contribution of our work is creating “counterfactual links” for improving ***graph representation learning*** and ***link prediction*** -- just like we put in the title of the submission: “___ Graph Learning for Link Prediction.” To the best of our knowledge, this is the ***first attempt*** to advance graph learning models and link prediction methods with the discovery from a causal model. We’d like to let the Graph Machine Learning community (or at least the Link Prediction community) know this new state-of-the-art method. A great number of NeurIPS organizers, participants, and paper readers are interested in these topics.
> > > > >
> > > > > Also, we are happy to share good news with the Causal Modeling community that even a basic causal model can significantly improve the performance of link prediction on a variety of graph datasets. Machine learning, or specifically graph representation learning, becomes one more research field that can benefit from the great line of work of Causal Modeling. It is time for the Graph Learning community to pay attention to these models, methods, and studies (after reading our paper which is the starting point, instead of the end, of the amazing journey).
> > > > >
> > > > > We want to take this chance to clarify the target task of this work, which is Link Prediction. It is a binary classification task - predicting binary labels of data objects. The data objects are node pairs. The labels describe whether there are links between the node pairs. Graph provides context (and structural features) of the node pairs. Therefore, the classification models make (multiple) observations from the (multiple) data objects, not one observation from the graph. Our task setting is standard. It is exactly the same as in the baseline methods. Evaluation methods are also standard. Causal modeling is used technically as an innovation of our link prediction method. Our method competed with the baselines in fair settings (in terms of datasets, computational resources, and evaluation methods).
> > > > >
> > > > > Our work delivered convincing experimental results showing that our innovative method outperformed very competitive baseline methods. So far, all reviews have reflected no need of adding new results towards a publication. We foresee applying significantly complex causal models as promising future work. The material you pointed out has lots of references that are worth consideration in the future work. Thank you again for your time and suggestions!

---

> > > > > > ### Author Response · Authors · 2021-08-28
> > > > > > **Response to Reviewer fY1x**
> > > > > >
> > > > > > We also want to point out that we did not claim robustness to a counterfactual change in this work.

---

### Official Review · Reviewer_9Usb · 2021-07-17

**Rating:** 7
**Confidence:** 4

**Summary:**

The authors propose a model for link prediction that takes into account the counterfactuals. The main idea is to learn an embedding of the nodes and then use that embedding to train a classifier model for link prediction. The key part is that the embedding is learned in such a way as to minimize the discrepancy between the factual and counterfactual distributions.
Experiments of various commonly-used datasets show that the proposed method results in improved link prediction performance compared to state-of-the-art methods.

**Limitations And Societal Impact:**

The authors adequately address the limitations of the work. Please, see above for a potential societal impact to address.

**Main Review:**

Originality: The proposed counterfactual learning framework is very similar to the work of Johansson et al., 2016, with the difference that the authors inserted a graph neural network with suitable. In my opinion, the innovative part of the work is considering various structural properties, such as belonging to the same community, as treatments.

Quality: The submission is technically sound. The proposed method builds naturally on existing work, and the experimental results show a clear advantage over state-of-the-art-methods. I have a concern when it comes to interpreting mutual community membership as a treatment. This raises the question of whether an adversarial party could decide to encourage nodes in the graph to connect / disconnect in order to manipulate the network for a malicious end.

Clarity: The paper is well-organized. Readers familiar with causal inference concepts as well as readers proficient with graph-based methods should be able to follow the paper and reproduce the results.

Significance: I believe the proposed work could be built upon by other researchers interested in causal learning in graphs. I would be interested to see follow up work on other problems besides link prediction.

**Time Spent Reviewing:**

2.5

---

> ### Author Response · Authors · 2021-08-09
> **Our response**
>
> Dear reviewer 9Usb, we appreciate your careful reading and insightful comments.
>
> Regarding the concern on the potential societal impact, thanks for raising this valuable concern. It is absolutely important to consider, study, and prevent adversarial attacks on the proposed graph learning model. We will discuss this point in the “Limitations” section. Techniques from related work could be employed or enhanced to address this limitation [1,2,3].
>
> [1] “Adversarial attack on graph structured data.” ICML 2018.
>
> [2] “Adversarial attacks on neural networks for graph data.” KDD 2018.
>
> [3] “GNNGuard: Defending Graph Neural Networks against Adversarial Attacks.” NeurIPS 2020.

---

### Decision · Program_Chairs · 2021-09-27

**Decision:**

Reject

**Comment:**

This paper proposes a new approach to link prediction problem inspired by causal inference. The idea of "counterfactual link" is very interesting and the effectiveness of the proposed method, which is expected to work well intuitively, is also well supported experimentally.
However, discussion of its theoretical background is somewhat lacking. While we agree that the central interest of this paper is not to give a causal model on graphs, a theoretical discussion of the conditions and assumptions under which the proposed method behaves well would strengthen the argument of the paper.
This paper is a solid contribution, but considering the competitive rate of NeurIPS, we have no choice but to give up the acceptance of this paper with the quality to be accepted at other conferences.